# Depressive symptoms exacerbate disability in older adults: A prospective cohort analysis of participants in the MemAID trial

Stephanie S. Buss[1]*, Laura Aponte Becerra[1¤a], Jorge Trevino[1¤b], Catherine B. Fortier[2,3], Long H. Ngo[4], Vera Novak[1]

**1** Department of Neurology, Beth Israel Deaconess Medical Center (BIDMC), Harvard Medical School Boston (HMS), Boston, Massachusetts, United States of America, **2** Translational Research Center for TBI and Stress Disorders (TRACTS) and Geriatric Research Educational and Clinical Research Center (GRECC) VA Boston Healthcare System, Boston, Massachusetts, United States of America, **3** Department of Psychiatry, Harvard Medical School Boston, Boston, Massachusetts, United States of America, **4** Department of Medicine, Beth Israel Deaconess Medical Center and School of Public Health, Harvard Medical School Boston, Boston, Massachusetts, United States of America

¤a Current address: Department of Internal Medicine, Jackson Memorial Hospital and University of Miami Miller School of Medicine, Miami, Florida, United States of America
¤b Current address: Division of Child Neurology, Children's Medical Center, UT Southwestern, Dallas, Texas, United States of America
* sbuss@bidmc.harvard.edu

**Data Availability Statement:** Data are available upon request due to ethical restrictions imposed by our study's IRB. The dataset from this study is property of Beth Israel Deaconess Medical Center

## Abstract

### Background

Maintaining independence in older age is an important aspect of quality of life. We investigated depressive symptoms as an important modifiable risk factor that may mediate the effects of physical and cognitive decline on disability.

### Methods

We prospectively analyzed data from 223 adults (age 50–85; 117 controls and 106 with type-2 diabetes) over 48 weeks who were participating in a clinical trial "Memory Advancement by Intranasal Insulin in Type 2 Diabetes." Data from self-reported disability (World Health Organization Disability Assessment Schedule) and depressive symptoms (Geriatric Depression Scale) were obtained from baseline, week 25, and week 48 visits. Cognition (Mini-mental status examination) and medical comorbidities (Charlson Comorbidity Index) were assessed at baseline. Longitudinal analysis assessed the extent to which change in depressive symptoms predicted worsening disability. Mediation analyses were performed to determine the extent to which depressive symptoms accounted for disability associated with worse cognition, walking speed, and comorbidities.

### Results

At baseline, depressive symptoms, cognition, and walking speed were within normal limits, but participants had a high 10-year risk of cardiovascular mortality. Depressive symptoms

(BIDMC) and contains sensitive medical information including information about past and current treatment of psychiatric disorders. In order to provide access to deidentified data from this dataset, BIDMC will require a data sharing agreement with a requesting investigator's institution. Additionally, the BIDMC IRB would require acknowledgment that the receiver has obtained an exemption from their local ethics/IRB committee that any shared data is exempt from Human Subject Research. Therefore, the data can be accessed by reaching out to the Research Administrator of the BIDMC Neurology Department, who would then put the necessary agreements in place to facilitate data sharing. Name: Stacy Mueller Title: Research Administrator, Neurology Department, BIDMC Email: slmuelle@bidmc.harvard.edu Phone: 617-667-1984 Address: Beth Israel Deaconess Medical Center, 330 Brookline Ave, Boston MA.

**Funding:** This work was supported by grants from the National Institutes of Health (NIDDK-1R01DK103902-5 to V.N.); Harvard Catalyst - The Harvard Clinical and Translational Science Center (National Center for Advancing Translational Sciences, National Institutes of Health Award UL 1TR002541); and financial contributions from Harvard University and its affiliated academic healthcare centers. The clinical trial from which this data was drawn corresponds to FDA IND 107690 and was further supported with study drug from Novo-Nordisk Inc.; Bagsværd, Denmark through an independent ISS grant (ISS-001063) (to V.N). A safety sub-study was supported with CGM monitoring devices and supplies from Medtronic Inc., Northridge CA, USA through an independent grant NERP15-031 (to V.N). S.S.B. was further supported by the National Institutes of Health (1K23AG068384-01A1), Sidney R. Baer Jr. Foundation (01028951), the Alzheimer's Association (2019-AACSF-643094), and NeuroNEXT (U24NS107183). Role of Funding Sources: None of the funding agencies contributed to study design, subject recruitment, data collection, or data analysis. Novo-Nordisk and Medtronic reviewed the manuscript and made minor comments which were incorporated into the final submission.

**Competing interests:** S.S.B. served as a consultant for Kinto Care from 2019-2020. L.N. provided consultation to the Radiological Society; to the Journal of Cardiovascular Magnetic Resonance; to Five Island Consulting LLC, Georgetown ME; and to Vinmec Inc. Hanoi, Vietnam between 2015 and 2020. This does not alter our adherence to PLOS ONE policies on sharing data and materials.

were related to disability at baseline ($p<0.001$), and longitudinally ($p<0.001$). Cognition, walking speed, and comorbidities were associated with disability at baseline (p-values = 0.027–0.001). Depressive symptoms had a large mediating effect on disability longitudinally: the indirect effect on disability via depression accounts for 51% of the effect of cognition, 34% of the effect of mobility, and 24% of the effect of comorbidities.

## Conclusions

Depressive symptoms substantially exacerbated the effects of worsening cognition, gait speed, and comorbidities on disability. In our sample, most individuals scored within the "normal" range of the Geriatric Depression Scale, suggesting that even subclinical symptoms can lead to disability. Treating subclinical depression, which may be under-recognized in older adults, should be a public health priority to help preserve independence with aging.

## Introduction

The population of adults over the age of 50 currently makes up 35 percent of the US population [1], making the preservation of independence in older adults a public health priority. The prevention of cognitive, physical, and medical disability in older adults is one important component of promoting active and independent living in older adults [2, 3].

Disability can be defined under a "bio-psycho-social model" as limitations in activities of daily living arising from the interactions between health conditions and environment [4]. Multiple medical and social factors increase the risk of disability in older adults including physical inactivity [5], slow walking [6], cognitive impairment [7], cardiovascular risk factors [8], diabetes [9], and depressive symptoms [7]. Depressive symptoms are found in 11–25.3 percent of older adults [10] and are independently associated with disability [11]. In particular, anxiety and somatic symptoms related to depression are linked with a greater risk of disability in older adults [12]. Furthermore, depression may exacerbate disability when comorbid with other conditions such as chronic pain [13]. However, it remains unknown whether depressive symptoms prospectively mediate the relationship between cognition, mobility, medical risk factors and disability.

To address these knowledge gaps, we studied a sample of community-dwelling older adults who were concurrently participating in the Memory Advancement with Intranasal Insulin (MemAID) clinical trial [14, 15]. We hypothesized that worsening depressive symptoms would predict increasing disability prospectively. We further predicted that depressive symptoms would strongly mediate the effects of cognition, gait speed, and medical comorbidities on disability.

## Methods and materials

### Study setting

All study procedures were conducted at the Syncope and Falls in the Elderly (SAFE) Laboratory at the Beth Israel Deaconess Medical Center (BIDMC) and Brigham and Women's Hospital (BWH) Clinical Research Centers.

**Abbreviations:** BIDMC, Beth Israel Deaconess Medical Center; BMI, body mass index; BWH, Brigham and Women's Hospital; CCI, Charlson Comorbidity Index; GDS, Geriatric Depression Scale; HbA1$_C$, hemoglobin A1c; HOMA-IR, homeostatic model assessment of insulin resistance; HTN, hypertension; ICD-10, International Classification of Diseases 10th edition; MemAID study, Memory Advancement with Intranasal Insulin study; MMSE, Mini-Mental State Examination; NW, Normal walking; SAFE Lab, Syncope and Falls in the Elderly Laboratory; T2DM, type-2 diabetes; WHODAS, World Health Organization Disability Assessment Schedule 2.0; WTAR, Wechsler Test of Adult Reading.

## Ethics approval

All participants signed a written informed consent after research procedures were explained. The study was approved by the BIDMC Institutional Review Board (2015P-000064), with a cede review from BWH (2015P-000064), in compliance with the Declaration of Helsinki.

## Study design

All participants in this prospective cohort were concurrently enrolled in the MemAID study, a randomized, double-blinded, placebo-controlled clinical trial (ClinicalTrials.gov NCT02415556, FDA IND 107690). All study procedures occurred between October 6, 2015 and May 31, 2020. MemAID methodology has been previously published [14]; S1 Fig shows the relationship of the present dataset to the overall MemAID trial. Briefly, study participants completed screening and baseline assessments prior to initiation of the study drug or placebo. These included a comprehensive medical history, physical examination, blood draw, MMSE, Geriatric Depression Scale (GDS), 36-item World Health Organization Disability Assessment Schedule 2.0 (WHODAS) questionnaire administration [16], and gait assessment. Participants were asked to self-report demographic information including sex (Male or Female), race (American Indian or Alaska Native, Asian, Black or African American, Native Hawaiian or Other Pacific Islander, White, or more than one race), ethnicity (Hispanic or Latino or Not Hispanic or Latino), and years of education (total years including graduate education if applicable). Participants then attended nine study visits over 48 weeks, which included assessment of GDS, WHODAS, and gait. In order to minimize confounding from any treatment effect, data for the present analysis was used only from time-points before or after study drug administration.

## Participants

Eligible participants were 50 to 85 years old, with or without type-2 diabetes (T2DM), able to walk for six minutes, had a Mini-Mental State Examination (MMSE) >20 with no diagnosed dementia, and had no major medical conditions or surgeries within the last six months. For the cross-sectional analysis, we used data from 223 participants who completed screening and baseline as part of the MemAID trial (117 controls and 106 with T2DM) who underwent comprehensive assessments of medical status, cognitive function, depressive symptoms, and disability. The longitudinal analysis was performed on 155 participants who completed the study including final WHODAS assessment at 48 weeks (Fig 1).

## Assessment of disability and functionality

The WHODAS 2.0 is a self-reported measure of disability. The WHODAS quantifies difficulty preforming activities of daily living in six functional domains: cognition, mobility, self-care, getting along with people, life activities, and participation in society [16]. For each participant, a WHODAS 2.0 Complex Score was calculated to represent the global disability of each participant on a scale from 0 to 100; with 0 representing no disability and 100 representing full disability [16]. Participants were further categorized based on WHODAS into Absent-Mild Disability (0–24 WHODAS score) and Moderate-Severe Disability (25–95 WHODAS score) groups [17].

## Assessment of depressive symptoms and cognitive function

The MMSE was used to evaluate cognition (range 0–30; score >24 indicates intact cognition) [18]. The Wechsler Test of Adult Reading (WTAR) was administered at baseline to estimate

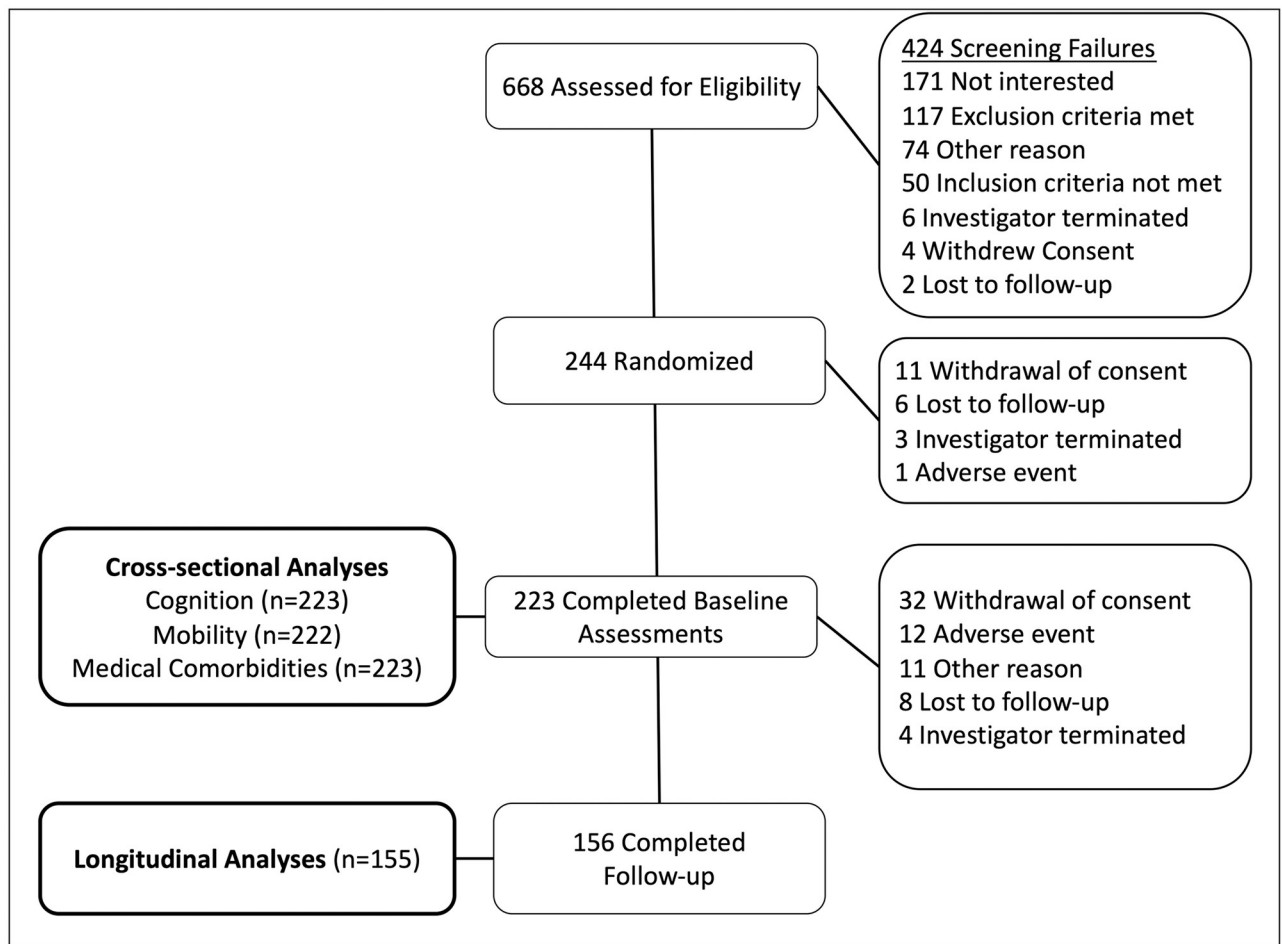

**Fig 1. CONSORT diagram.** Participant flow of the Memory Advancement with Intranasal insulin (MemAID) participants and relationship with this prospective cohort study. All 223 participants who completed screening and baseline assessments were included in the cross-sectional analyses. Of these, 66 did not complete the MemAID study (either withdrew from the study, were terminated by the investigator, or were lost to follow-up), and one completed the MemAID study but was missing data for the week 48 WHODAS. Therefore, the remaining 155 participants were included in the longitudinal analysis of this prospective cohort.

premorbid intellectual functioning (standard score mean = 100; SD = 15 corresponding with age-adjusted IQ) [19]. The Geriatric Depression Scale was used to measure self-reported depressive symptoms over the past week (GDS; range 0–30; 0–9 normal; 10–19 mild, 20–30 severe) [20].

## Assessment of mobility

Walking was measured by the Mobility Lab System (APDM, Inc., Portland, OR.) during six minutes of normal walking (NW) in 45 m hallway. Gait speed was measured in centimeters per second (cm/sec), excluding turns. NW speed predicts longer survival in adults over 65 years (normal range 80–150 cm/sec) [6].

## Assessment of medical comorbidities and 10-year mortality risk

The Charlson Comorbidity Index (CCI) is a validated prognostic mortality indicator and measure of disease burden [21] which is based on the International Classification of Diseases 10th

edition (ICD-10) coding. The CCI is widely used to estimate mortality risk from multiple comorbidities [22]. CCI-based scores are related to 10-year mortality among hospitalized patients: a score of 0 = 8% risk, a score of 1 = 25% risk, a score of 2 = 48% risk, and a score of 3 = 59% risk [22]. However, actual cardiovascular risk estimates are likely lower in our cohort due to advances in medical care since the original CCI validation and since participants were evaluated outpatient.

## Statistical analyses

JMP Pro 15.0 (SAS Institute Inc., Cary, NC) was used for statistical analysis. The clinical dependent variable of interest was WHODAS. The independent variables of interest were NW, MMSE, and CCI, and the independent variable investigated as a mediator was GDS.

Baseline characteristics are reported in the Table 1 for the whole sample, as well as for the Absent-Mild and Moderate-Severe Disability subgroups.

**Power calculation.** With a power at 0.8 or higher and a type 1 error set at 0.05, our sample of 223 participants in the cross-sectional analysis allows us to detect an effect size of r = 0.19 (correlation coefficient; a moderate association) for the association between each of the three independent variables of interest (MMSE, NW, CCI) and the dependent variable of interest (WHODAS).

In the longitudinal analysis, our sample of 155 participants allows us to detect an effect size of r = 0.22 (a moderate association) for the association between each of the three independent variables of interest (MMSE, NW, CCI) and the dependent variable of interest (WHODAS) with a power at 0.8 or higher and a type 1 error set at 0.05.

**Cross-sectional analyses.** All 223 participants who completed baseline assessments were included in cross-sectional analyses. Four separate fixed-effects linear model tested the relationship between each measure of functionality (GDS, MMSE, NW, and CCI; independent variables) and WHODAS (the dependent variable). Covariates for the GDS, MMSE, and NW models were age, sex, education, hemoglobin A1c (HbA1$_C$), and race. For the CCI model, only education and race were included as covariates, since age, sex, and history of T2DM were already included in the CCI score. Race was encoded as a dichotomous variable (White vs. Non-White).

**Longitudinal analysis of change in disability.** To clarify a possible causal relationship between depressive symptoms and disability, we examined how early changes in GDS predicted later change in WHODAS in the 155 participants who completed longitudinal assessment of disability.

To examine how GDS changed over the course of the study, a Tukey-Kramer HSD test was used to compare mean GDS scores at baseline, at the first post-treatment visit (25 weeks; Mid-study GDS), and at the final study visit (48-week GDS).

WHODAS Change was calculated as difference between WHODAS at the final study visit (48-week WHODAS) and WHODAS score at baseline. GDS Change was calculated as the difference between Mid-study GDS and baseline GDS. Nine subjects who completed 48-week WHODAS assessment were missing Mid-study GDS data; for these participants data was imputed using last-value carried forward from their prior study visit (approximately 165 days from baseline). These visits were chosen since they were outside the treatment period of intranasal insulin to minimize any potential treatment effects.

A fixed-effects linear model assessed the relationship between GDS Change and WHODAS Change. Covariates of age, sex, education, HbA1$_C$, treatment group (i.e., exposure to intranasal insulin), and race were included in the model.

**Table 1. Demographics and clinical variables.**

| Demographics and clinical variables at baseline | All Participants (n = 223) | Absent-Mild Disability (n = 192) | Moderate-Severe Disability (n = 31) | p-value |
|---|---|---|---|---|
| Age[a] | 65.6 ± 9.0 | 66.2 ± 9.1 | 62.0 ± 8.1 | 0.016 |
| Sex (% Female)[b] | 48.9% | 49.5% | 45.2% | 0.702 |
| Race[c] | | | | <0.001 |
| % White | 77.6% | 82.3% | 48.4% | |
| % Black or African-American | 15.2% | 12.0% | 35.5% | |
| % Asian | 3.6% | 3.6% | 3.2% | |
| % More than one race | 3.1% | 1.6% | 12.9% | |
| % Unknown | 0.4% | 0.5% | 0% | |
| Ethnicity (% Latinx)[b] | 5.8% | 4.2% | 16.1% | 0.022 |
| Years of Education[a] | 16.3 ± 3.4 | 16.5 ± 3.3 | 14.9 ± 3.6 | 0.019 |
| Employment (% Employed)[b] | 35.4% | 38.5% | 16.1% | 0.015 |
| History of T2DM (%)[b] | 47.5% | 41.2% | 87.1% | <0.001 |
| History of HTN (%)[b] | 48.0% | 44.3% | 71.0% | 0.007 |
| History of Mental Illness (%)[b] | 27.8% | 25.0% | 45.2% | 0.030 |
| History of Depression (%)[b] | 18.4% | 17.2% | 25.8% | 0.315 |
| BMI[a] | 29.5 ± 6.2 | 28.9 ± 5.8 | 33.2 ± 7.5 | <0.001 |
| HOMA1-IR[a] | 4.1 ± 5.7 | 3.6 ± 4.3 | 7.5 ± 10.5 | <0.001 |
| Fasting glucose[a] | 114.4 ± 41.4 | 112.1 ± 40.2 | 129.1 ± 45.8 | 0.034 |
| HbA1$_C$[a] | 6.3 ± 1.3 | 6.2 ± 1.2 | 7.1 ± 1.5 | <0.001 |
| # of Medications[a] | 7.5 ± 6.3 | 6.9 ± 6.1 | 11.3 ± 5.9 | <0.001 |
| Medication for Depression (%)[b] | 19.5% | 18.3% | 26.7% | 0.320 |
| Medication for Diabetes (%)[b] | 39.5% | 35.6% | 73.3% | <0.001 |
| WTAR (age-adjusted IQ)[a] | 112.7 ± 13.7 | 114.3 ±12.6 | 103.3 ± 16.6 | <0.001 |
| MMSE[a] | 28.3 ± 1.8 | 28.4 ± 1.7 | 27.6 ± 2.3 | 0.015 |
| GDS[a] | 5.7 ± 5.3 | 4.5 ± 4.4 | 12.7 ± 5.1 | <0.001 |
| WHODAS[a] | 12.0 ± 12.2 | 8.0 ± 6.9 | 36.3 ± 9.1 | <0.001 |
| CCI[a] | 3.4 ± 1.7 | 3.3 ± 1.7 | 3.5 ± 1.4 | 0.498 |
| Normal Walking Speed (cm/sec)[a] | 114.4 ± 21.5 | 116.0 ± 20.4 | 104.4 ± 25.8 | 0.005 |

Key:

[a]Pooled 2-tailed t-test assuming equal variance

[b]2-Tail Fisher's Exact test

[c]Pearson's Chi-square test

Demographic and clinical characteristics are reported as Mean ± SD for continuous variables or % for categorical variables. The p-value indicates a difference in mean between the Absent-Mild Disability subgroup and the Moderate-Severe Disability subgroup as defined by WHODAS score. Thirteen participants had missing data on medications and one patient was missing information on NW Gait Speed at baseline. Demographics shown include self-reported variables of sex (male/female), race and ethnicity [23], and years of education. T2DM = type-2 diabetes, HTN = hypertension, BMI = body mass index, HOMA-IR = homeostatic model assessment of insulin resistance, HbA1$_C$ = hemoglobin A1c, WTAR = Weschler test of adult reading, MMSE = mini-mental state exam, GDS = geriatric depression scale, WHODAS = WHO Assessment Schedule 2.0, CCI = Charlson comorbidity index.

**Longitudinal mediation analyses.** Data for the longitudinal mediation analysis were included from 155 participants who completed 48-week WHODAS. One participant who completed the MemAID trial did not complete the 48-week WHODAS so was excluded from the longitudinal analysis. The dependent variable was 48-week WHODAS (approximately 333 days from baseline assessments). Mid-study GDS (approximately 173 days from baseline assessments) was used a covariate for the mediation analysis.

Three separate fixed-effects linear models tested for the effect of each independent variable at baseline (MMSE, NW, and CCI) on 48-week WHODAS (dependent variable). Covariates

for the MMSE and NW models were age, sex, education, $HbA1_C$, treatment group, and race. For the CCI model education, treatment group, and race were included as covariates.

To test for mediation effects, Mid-study GDS at 25 weeks was added to each linear model as the potential mediating variable. For each independent variable, the Total effect was calculated as the B coefficient of the independent variable on 48-week WHODAS. Separate models then tested the effect of each independent variable on Mid-study GDS, and of Mid-study GDS on 48-week WHODAS. The Indirect effect of each independent variable was calculated as the B coefficient of the independent variable on Mid-study GDS multiplied by the B coefficient of Mid-study GDS on 48-week WHODAS. A Direct effect of each independent variable was calculated by subtracting the Indirect effect from the Total effect, and the % of Direct and Indirect contribution to the Total effect was calculated (e.g., % Direct Effect = (Direct Effect/Total Effect)*100).

To clarify if mediation effects were already present at baseline, a complementary cross-sectional mediation analysis was also performed (S1 Appendix: Methods).

**Post-hoc T2DM subgroup analysis.**   Given the pervasive significant effect of $HbA1_C$ across multiple models, we ran a post-hoc analysis within the diabetes subgroup test whether disease severity or treatment affected disability (S1 Appendix: Methods).

## Results

### Demographic and clinical variables

Table 1 shows characteristics of the 223 participants (117 controls; 106 T2DM); GDS (5.7 ±5.3), MMSE (28.3±1.8), and NW speed (114.4±21.5 cm/s) were within normal limits. The majority of participants (175) did not show clinically significant depression; 44 participants had mild depressive symptoms and four participants had severe depressive symptoms. The majority (78%) of participants self-reported race as "White."

Compared to participants with Absent-Mild Disability, participants with Moderate-Severe Disability were more likely to be younger, had fewer years of education, and had a greater diversity of racial/ethnic background, with a greater proportion of participants identifying as Black/African-American, multi-racial, and Latinx (p-values <0.001 to 0.019). Patients with Moderate-Severe Disability had worse health status across a number of health indicators (including NW, T2DM, HTN, BMI, $HbA1_C$, employment, mental illness, and IQ), although CCI was not different between groups (Table 1). Participants with Moderate-Severe Disability had clinically significant depression on GDS (p<0.001, with a mean of 12.7 indicating mild depression) compared to those with mild disability. However, treatment with medications for depression was comparable to the Absent-Mild Disability group, indicating that they were not more likely to receive pharmacologic therapy despite the greater severity of depressive symptoms.

### Cross-sectional analyses

A greater GDS score at baseline was associated with higher disability score on WHODAS ($R^2_{adj}$ = 0.47, d.f. = 222, B = 1.35, $p<0.001$, Fig 2a). Higher $HbA1_C$ (B = 1.59, $p$ = 0.002), Non-White race (B = 1.60, $p$ = 0.043) and lower education (B = -0.40, $p$ = 0.029) were significant covariates associated with higher WHODAS.

Lower MMSE was related to higher WHODAS ($R^2_{adj}$ = 0.15, d.f. = 222, B = -1.01, $p$ = 0.027, Fig 2b and S1 Appendix). Higher $HbA1_C$ (B = 2.43, $p<0.001$) and Non-White race (B = 2.07, $p$ = 0.039) were related to higher WHODAS.

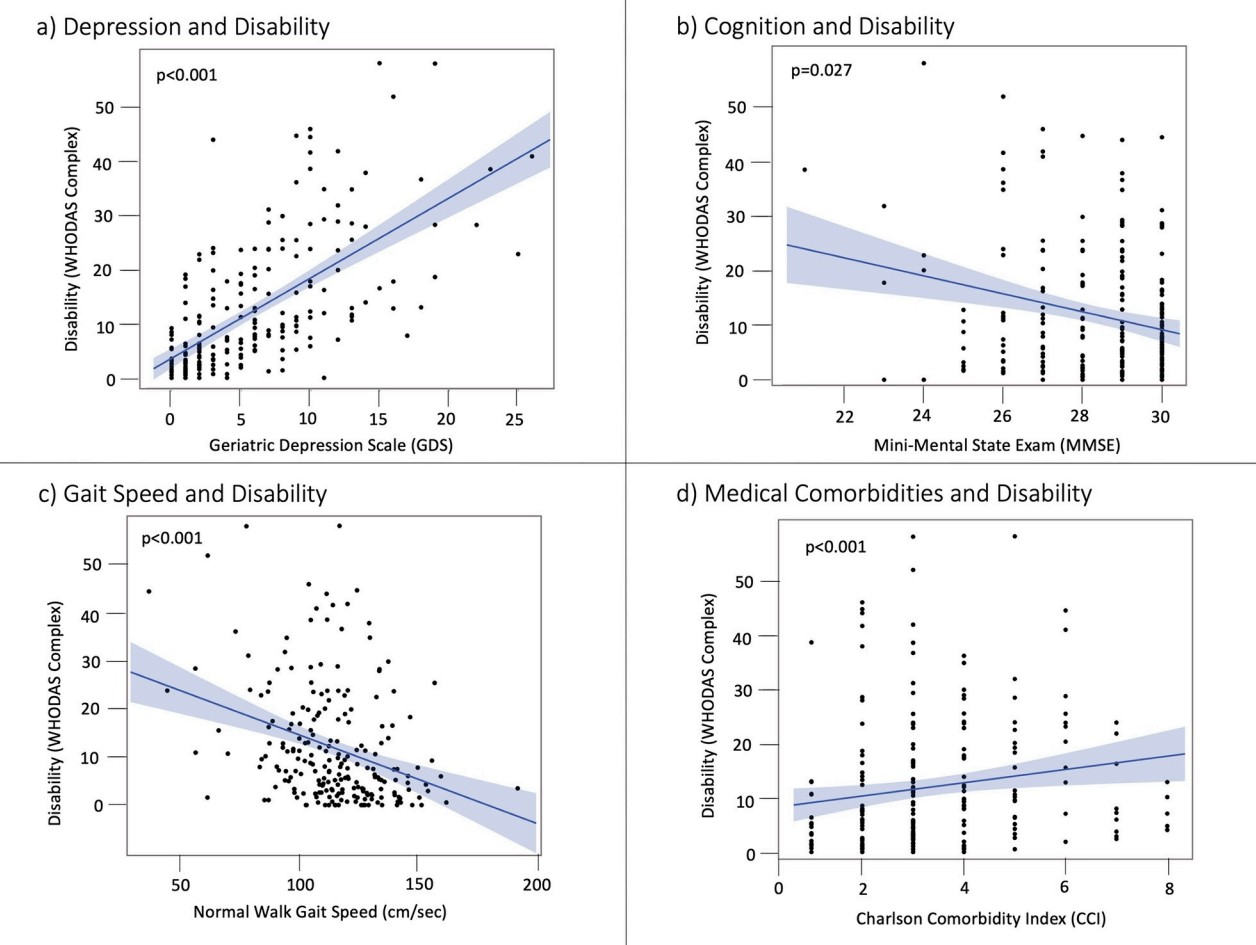

**Fig 2. Association between cognition, gait and comorbidities and disability at baseline. a:** Greater disability on the World Health Organization Disability Assessment Schedule 2.0 (WHODAS) was associated with higher depression scores on the Geriatric Depression Scale (GDS, $p<0.001$). **b:** More disability was associated with worse cognition on Mini-Mental State Examination (MMSE, $p = 0.027$). **c:** More disability was associated with slower gait speed during normal walk (NW, $p<0.001$). **d:** More disability was associated with medical comorbidities on the Charlson Comorbidity Index total points (CCI, $p<0.001$).

Slower NW was associated with greater disability on WHODAS ($R^2_{adj}$ = 0.19, d.f. = 221, B = -0.14, $p<0.001$, Fig 2c and S1 Table in S1 Appendix). Covariates of higher HbA1$_C$ (B = 2.05, $p$ = 0.001) and Non-White race (B = 1.98, $p$ = 0.043) were also related to higher WHODAS.

Higher CCI was correlated with higher WHODAS ($R^2_{adj}$ = 0.14, d.f. = 222, B = 1.77, $p<0.001$, Fig 2d and S1 Appendix). Lower education (B = -0.70, $p$ = 0.003) and Non-White race (B = 3.60, $p<0.001$) were related to higher WHODAS.

## Change in disability and depression over 48 weeks

At the group level, there was no significant difference in GDS at baseline, Mid-Study, or 48 weeks (S2 Fig). However, while mean WHODAS and GDS were relatively stable, there was clinically meaningful variability within individual participants during the course of the study in WHODAS Change (mean = -1.3, S.D. = 8.5, range -42.3 to 31.5) and GDS Change (mean = -0.45, S.D. = 4.0, range -12 to 21). Of the participants without depressive symptoms at baseline, 12 developed depressive symptoms by Mid-Study (GDS>10). Overall during the course of the

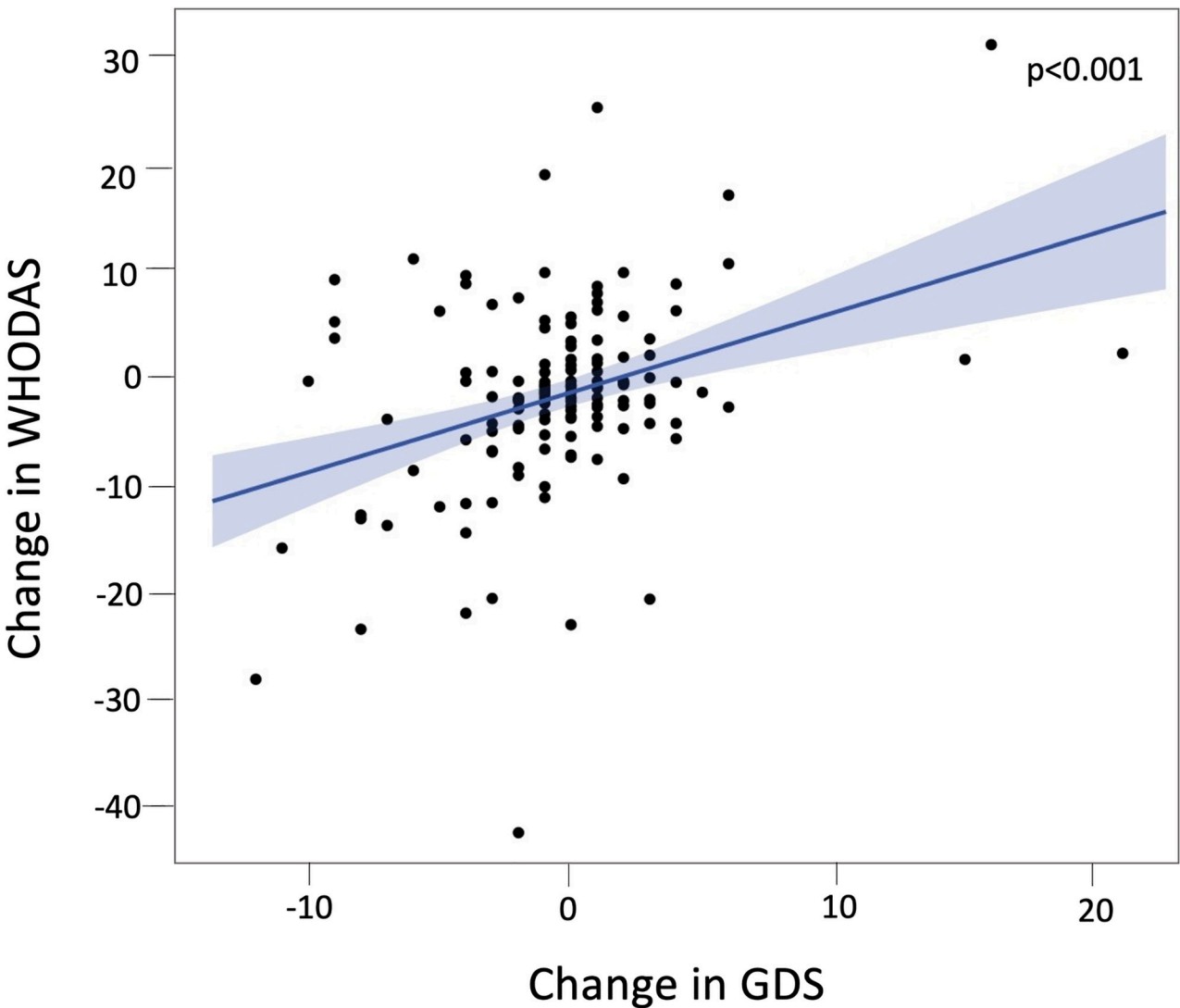

**Fig 3. Change in depressive symptoms predicts change in disability over 48 weeks.** Worsening depressive symptoms on GDS (baseline to Mid-Study) were related to increasing disability on WHODAS (baseline to week 48) ($p < 0.001$). Forty-six participants experienced worsening of both GDS and WHODAS, while 24 participants experienced improvement on both measures. A 10-point increase in GDS corresponds to conversion from no depression to mild depression. WHODAS 2.0 Complex score ranges from 0 to 100, with each 10-point increasing corresponding to a 10% increase in overall disability.

trial, 46 participants experienced both worsening GDS and worsening WHODAS, while 24 participants experienced both improving GDS and improving WHODAS. Greater GDS Change was associated with higher WHODAS Change ($R^2_{adj} = 0.10$, d.f. = 154, B = 0.72, $p < 0.001$) when accounting for covariates (Fig 3). This effect size was clinically significant: a 10-point increase in GDS (corresponding to a conversion from no depression to mild depression) was associated with a seven-point increase in WHODAS (corresponding to 7% greater disability).

### Longitudinal mediation analyses

Lower MMSE at baseline was related to a higher 48-week WHODAS ($R^2_{adj} = 0.15$, d.f. = 154, B = -1.51, $p = 0.006$); higher HbA1$_C$ was also related to WHODAS (B = 2.81, $p = 0.001$). When

**Table 2. Mediation effect of depression on the relationship between gait speed, cognition, and medical comorbidities on disability.**

| Measure of Functionality | Primary Model | | | Model with GDS | | | Change in B | Direct Effect | Indirect Effect |
|---|---|---|---|---|---|---|---|---|---|
| | B | d.f. | p-value | B | d.f. | p-value | % change | % of Total | % of Total |
| Cognition (MMSE) | -1.51 | 154 | 0.006 | -0.77 | 154 | 0.092 | 49.0% | 49.4% | 50.6% |
| Gait Speed (NW) | -0.13 | 153 | 0.005 | -0.08 | 153 | 0.021 | 38.5% | 65.8% | 34.2% |
| Comorbidities (CCI) | 2.22 | 154 | <0.001 | 1.71 | 154 | <0.001 | 23.0% | 75.7% | 24.3% |

Mid-study GDS was added to the model, MMSE was no longer a significant variable (B = -0.77, $p$ = 0.092). Mid-study GDS was a partial mediator of the effect of MMSE on 48-week WHODAS, with an Indirect effect of 50.6% of the Total (Table 2 and Fig 4a).

**Coefficients of longitudinal mediation analyses.** The effect size of each independent variable (NW, MMSE, and CCI) on disability is shown for each model used in the mediation

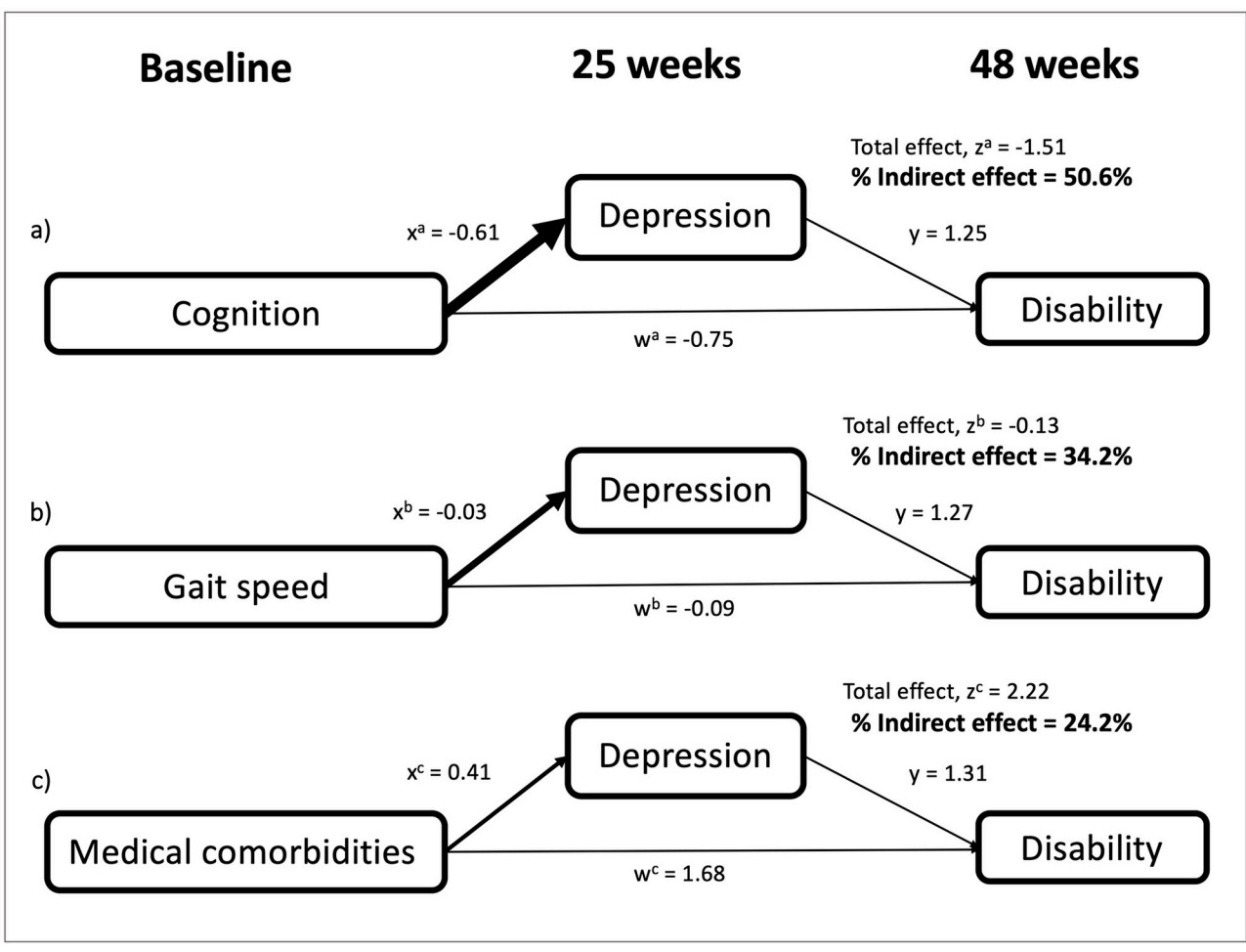

**Fig 4. Longitudinal mediation analysis models.** A mediation analysis was performed to measure the extent to which depressive symptoms on the Geriatric depression scale (GDS) exacerbate the effects of (a) cognition, (b) gait speed, and (c) medical comorbidities on disability over 48 weeks. To investigate possible causality, the models used values from each independent variable at baseline, depressive symptoms at 25 weeks, and disability at 48 weeks. Coefficients for Direct pathways (w) and Indirect pathways (x and y) are shown. The "w" pathway represents the direct effect of each independent variable on disability, independent from the effect attributable to mediation by GDS. The "x" pathway represents the effect of each independent variable on GDS. The "y" pathway represents the effect of GDS on disability. The "z" coefficient represents the total effect, including both the Direct and Indirect pathways. The percent Indirect effect was calculated as % Indirect effect = (x*y)/z. Depressive symptoms accounted for 24–51% of the effect of each independent variable on disability in these longitudinal models.

analysis. A change in B > 10% after GDS is added to the model supports a mediation effect of GDS. The Direct effect represents the proportion of the effect of each independent variable on disability, independent of GDS. The Indirect effect represents the proportion of the total effect attributable to mediation by GDS.

Slower NW at baseline was associated with greater disability on 48-week WHODAS ($R^2_{adj}$ = 0.15, d.f. = 153, B = -0.13, $p$ = 0.005). Higher HbA1$_C$ was also related to WHODAS (B = 2.87, $p$ = 0.001). Mid-study GDS was a partial mediator of the effect of NW on 48-week WHODAS, with an Indirect effect of 34.2% of the Total (Table 2 and Fig 4b).

Higher CCI at baseline was correlated with higher 48-week WHODAS ($R^2_{adj}$ = 0.13, d.f. = 154, B = 2.22, $p$<0.001). Race (B = 3.13, $p$ = 0.007) and education (B = -0.59, $p$ = 0.034) were significant covariates. Mid-study GDS was a partial mediator of the effect of CCI on 48-week WHODAS, with an Indirect effect of 24.3% of the Total (Table 2 and Fig 4c).

## Cross-sectional mediation analyses

The cross-sectional mediation analysis showed results consistent with the longitudinal mediation analysis. Indirect effects ranged from 35–50% (S1 Appendix).

## Post-hoc T2DM subgroup analysis

In participants with T2DM, severity and treatment of T2DM was not associated with disability (S1 Appendix).

## Discussion

Our results show that depressive symptoms substantially exacerbate the effects of cognitive function, walking speed, and medical comorbidities on disability in older adults. In line with prior literature, worsening depressive symptoms predicted increasing disability over 48 weeks. Longitudinal mediation models showed that depressive symptoms mediated 51% of the effect of cognition, 34% of the effect of mobility, and 24% of the effect of medical comorbidities on disability. These large mediation effects were found in our sample of ambulatory community-dwelling adults who had walking speeds, cognitive function, and depressive symptoms largely within the normal ranges. Therefore, even mild depressive symptoms, often below the standard treatment threshold, predict worsening disability longitudinally. These results suggest that older adults are more vulnerable to the effects of depression, which may be underdiagnosed and/or undertreated. Targeted screening and treatment for depressive symptoms in at-risk older adults may help to reduce medical care costs and prevent disability. Future interventions to promote maintenance of independence in community-dwelling older adults will require integrative, multimodal interventions addressing physical function, cognition, lifestyle factors, medical comorbidities, along with mental health.

We found a strong relationship between depressive symptoms and disability. This finding is in line with prior literature showing that patients with late-onset depression are more likely to have comorbid medical conditions, poor physical function, and cognitive decline [11]. This relationship was found both cross-sectionally and longitudinally, with worsening depressive symptoms predicting subsequent increasing disability. Importantly, the average GDS in our sample fell within the "normal" range yet was still strongly associated with disability, indicating the strong effect of subclinical depressive symptoms. While major depressive disorder has a prevalence of 2% in older adults, subclinical depressive symptoms have been found to occur in 10–15% of older adults and are similarly associated with poor health outcomes [24, 25]. Our findings extend prior literature showing that older adults with subclinical depression have greater functional impairment than non-depressed adults [24]. Subclinical depression has

been linked to low socio-economic status, poor physical function, cognitive impairment, and low functional status, similar to the effects of clinical depression in older adults [24].

Cognitive impairment due to neurodegenerative disorders is a significant cause of disability worldwide [26]. Older adults who do not meet formal criteria for mild cognitive impairment (MCI) or dementia may nevertheless show cognitive decline with advancing age, which is often caused by the early stages of neuropathology [27]. Our finding that depression mediates the relationship between MMSE and disability highlights the importance of early intervention for both depressive symptoms and cognition in older adults with cognitive impairments. However, this relationship is complex and likely bi-directional since depression can lead to executive function difficulties [28] and early stage neurodegenerative disorders can lead to depressive symptoms and other neuropsychiatric symptoms [29].

Gait speed is an independent predictor of survival in adults over age 65, and walking slower than 80 cm/s increases the risk of early mortality [6]. Slower gait speed is also linked to disability, frailty, falls, sedentary lifestyle [30], stress and lower quality of life [31]. Slow gait and depressive symptoms may co-occur, leading to greater risk of incident disability than either factor alone [32]. Indeed, the association between gait speed and mortality seems to be strongest in patients with more severe depressive symptoms [33]. Our present results unify prior findings showing that slow gait speed is associated with development of depression and disability, and further support that depression plays a substantial mediating role between these factors.

Medical comorbidities are established strong predictors of disability and long-term health [34]. Our results show that depressive symptoms partially mediate this effect. Higher HbA1$_C$ was a strong predictor of disability across our models, suggesting a strong relationship between prediabetes, T2DM, and disability. Indeed, patients with T2DM have an approximately 2-fold increase in the prevalence of depression [9], and depression may make it more difficult for patients with T2DM to achieve glycemic control [35]. Diabetes is also linked to lower total brain volume, executive dysfunction, and accelerated rate of cognitive decline [36, 37]. Future research is needed to determine best practices for screening for patients with comorbid depression and diabetes to improve long-term health outcomes and level of independence.

Self-reported race was a significant covariate in several of our analyses, with minority status associated with greater disability over time. This finding mirrors literature showing that significant health disparities exist for adults with racial and ethnic minority backgrounds, conferring higher risk for a broad range of adverse health outcomes [38]. Therefore, future research on reducing disability must also be coupled with broader efforts to reduce healthcare disparities [39]. For example, it is important to develop and study service delivery models that can effectively reach patient populations with diverse backgrounds, create culturally appropriate interventions and educational materials, and promote equal access to healthcare and social support systems for people of all racial and ethnic backgrounds [38, 39].

Overall, our findings highlight the importance of preventing, detecting, and treating depressive symptoms in older adults, including subclinical depression. Primary prevention techniques for depression include exercise programs, relaxation techniques, cognitive restructuring, mind-body programs, social engagement, and sleep hygiene [11]. Treatment of clinical depression in older adults follows the same principles as in younger adults and is similarly effective, including psychotherapy, physical activity, and antidepressant medications [11, 25]. However, depression is often under-treated in older adults [25]. Under-diagnosis may be related to non-specific symptom presentation common in older adults including symptoms of fatigue, social withdrawal, or weight loss [25]. Comorbidity of depressive symptoms with medical and cognitive disorders, and concerns about medication side effects, may also complicate diagnosis and treatment of depression in older populations [25].

Given the complex interplay between depression, medical comorbidities, and physical function, efforts to reduce depressive symptoms in community-dwelling older adults will likely need to be integrated into multimodal interventions. For example, the large-scale FINGER study found that an intervention of diet modification, physical activity, cognitive training, and vascular risk factor management improved cognitive functioning, dietary habits, BMI, and physical activity levels in at-risk older adults [40]. However, since depressive symptoms may interfere with motivation, lifestyle interventions may need to be paired with motivational strategies, psychotherapy, or anti-depressant medications to be effective for adults with depression. In fact, targeted treatment of depression has shown evidence of improving cognitive outcomes and overall disability [41]. These data combined with our results suggest that assessment and treatment of subclinical depression should be an integral factor in future multidomain interventions in older adults.

Strengths of this study include a participant sample drawn from community-dwelling older adults, supporting the generalizability of our findings. The long follow-up (48 weeks) allows us to determine relationships with disability longitudinally and supports the causality of our mediation models. Participants had high levels of medical comorbidities, allowing us to investigate multiple contributors to disability in this representative community sample. In-depth medical history and standardized gait assessment were performed. Disability (WHODAS) and depressive symptoms (GDS) were assessed repeatedly and rigorously during the study period, allowing for analysis of longitudinal trajectories.

Limitations of this study included reliance on self-report and lack of formal psychiatric assessment or diagnosis by a psychiatric clinician. Depressive symptoms were assessed using GDS, and medical history of diagnosis or treatment of depression was provided by participant self-report. Structured psychological interviews were not preformed, limiting our ability to evaluate for the duration of any subtle subclinical depressive symptoms reflected in GDS score but not in clinical history. However, we do not think this effects the validity of our results, as has good sensitivity and specificity for the diagnosis of depression (92 and 89%, respectively) and provides an objective metric used as a screening tool in clinical practice [42]. In fact, since this study focused on community-dwelling older adults scoring largely in the "normal" range of depressive symptoms, we might expect that our results would be even stronger in other populations such as older adults with longstanding psychiatric disorders, or in more frail individuals living in institutional settings. It must also be noted that while our results suggest a strong mediating effect of depression, we are unable to draw firm causal conclusions based on the observational nature of the data. Future studies could examine biomarker-defined cohorts of older adults to examine whether treating depressive symptoms could prospectively could reduce the burden of disability in older adults.

## Conclusions

Our mediation analysis implicates depressive symptoms as a causal factor leading to disability in older adults: depressive symptoms substantially exacerbated the effects of cognition, gait speed and medical comorbidities. Large mediation effects were found in our sample of ambulatory older adults, despite most individuals score within the "normal" range for depressive symptoms. Therefore, subclinical depression should be specifically targeted when designing multifaceted public health interventions to reduce medical and societal costs and preserve independence in older adults.

## Supporting information

**S1 Fig. Study flow diagram.** All participants included in the present study were concurrently enrolled in the MemAID clinical trial of intranasal insulin. Only data included in the

longitudinal analysis are shown and were drawn from visits at baseline, 25 weeks, and 48 weeks. Additional assessments which were performed as part of the MemAID trial have been previously published (Novak, et al., Journal of Neurology 2022) and are not shown in the figure. INI: intranasal insulin; MMSE: Mini mental state examination; WHODAS: World Health Organization Disability Assessment Schedule 2.0; GDS: Geriatric Depression Scale.
(TIF)

**S2 Fig. Longitudinal depressive symptoms.** Mean depressive symptoms over the duration of the study are shown. At the group level, depressive symptoms were stable between baseline, Mid-study (week 25), and end of the study (week 48). GDS: Geriatric Depression Scale.
(TIF)

**S1 Appendix. Cross-sectional mediation analysis.** Evaluation of baseline data showed that the mediation effects seen in the longitudinal analysis were already present at baseline. **Post-hoc T2DM Subgroup analysis.** The group of participants with T2DM showed more disability than controls, however, markers of diabetes severity were not related to greater disability.
(DOCX)

## Acknowledgments

The authors thank the MemAID Investigators: Vasileios A. Lioutas MD, Peter Novak, MD, PhD, Regina E. McGlinchey, PhD (Site PI) for their contributions, dedicated time and skills for completion of the MemAID trial.

## Author Contributions

**Conceptualization:** Stephanie S. Buss, Laura Aponte Becerra, Jorge Trevino, Long H. Ngo, Vera Novak.

**Data curation:** Stephanie S. Buss, Laura Aponte Becerra, Jorge Trevino, Vera Novak.

**Formal analysis:** Stephanie S. Buss, Laura Aponte Becerra, Jorge Trevino, Long H. Ngo, Vera Novak.

**Funding acquisition:** Vera Novak.

**Investigation:** Laura Aponte Becerra, Jorge Trevino, Catherine B. Fortier, Vera Novak.

**Methodology:** Vera Novak.

**Project administration:** Laura Aponte Becerra, Jorge Trevino, Vera Novak.

**Resources:** Vera Novak.

**Software:** Vera Novak.

**Supervision:** Vera Novak.

**Validation:** Long H. Ngo.

**Visualization:** Stephanie S. Buss.

**Writing – original draft:** Stephanie S. Buss.

**Writing – review & editing:** Stephanie S. Buss, Laura Aponte Becerra, Jorge Trevino, Catherine B. Fortier, Long H. Ngo, Vera Novak.

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
