## [Decision Letter · Decision Letter 0]

9 Aug 2022

PONE-D-22-17003Depressive symptoms exacerbate disability in older adults:

A longitudinal cohort studyPLOS ONE

Dear Dr. Buss,

Thank you for submitting your manuscript to PLOS ONE. After careful consideration, we feel that it has merit but does not fully meet PLOS ONE’s publication criteria as it currently stands. Therefore, we invite you to submit a revised version of the manuscript that addresses the points raised during the review process.

Remark: Some formatting amendments are mandatory.               Please detail the methodology as appropriate.

We look forward to receiving your revised manuscript.

Kind regards,

Lai Kuan Lee

Academic Editor

PLOS ONE

Journal Requirements:

"S.S.B. served as a consultant for Kinto Care from 2019-2020. L.N. provided consultation to the Radiological Society; to the Journal of Cardiovascular Magnetic Resonance; to Five Island Consulting LLC, Georgetown ME; and to Vinmec Inc. Hanoi, Vietnam between 2015 and 2020. The authors report no conflicts with any product mentioned or concept discussed in this article."

Reviewers' comments:

Reviewer's Responses to Questions

**Comments to the Author**

1. Is the manuscript technically sound, and do the data support the conclusions?

Reviewer #1: Yes

Reviewer #2: Yes

2. Has the statistical analysis been performed appropriately and rigorously? 

Reviewer #1: Yes

Reviewer #2: I Don't Know

3. Have the authors made all data underlying the findings in their manuscript fully available?

Reviewer #1: No

Reviewer #2: No

4. Is the manuscript presented in an intelligible fashion and written in standard English?

Reviewer #1: No

Reviewer #2: Yes

5. Review Comments to the Author

Reviewer #1: Dear authors! Thank you for conducting this study, which might also assist mental health professionals in recognizing and treating depression symptoms in the elderly population to reduce disabilities related to depression symptoms . However, your methods and materials weren't quite clear. I've stated my review comments below, and each one should be addressed as appropriate.

Reviewer #2: This study has the merit as it is done on longitudinal basis. The whole study was described in an intelligent fashion, well structured methodology. From my point of view, the article can be accepted.

6. PLOS authors have the option to publish the peer review history of their article (what does this mean?). If published, this will include your full peer review and any attached files.

Reviewer #1: No

Reviewer #2: **Yes: **Dr. Panchanan Acharjee

---

## [Author Response · Author response to Decision Letter 0]

19 Oct 2022

Dear PLOS ONE Reviewers and Editorial Board,

We are pleased to submit a substantial revision of our manuscript (PONE-D-22-17003; new title “Depressive Symptoms exacerbate disability in older adults: a prospective cohort analysis of participants in the MemAID trial”). We thank the editors for the opportunity to resubmit and we thank the reviewers for their overall enthusiasm and detailed comments and suggestions for improvement. We believe the manuscript is much stronger as a result of their insights and the additional work we have completed to fully address their questions and comments. 

Changes in the manuscript are shown with tracked changes (marked-up copy). Briefly, as suggested by the reviewers’ comments and to address their questions, we have (1) described our research methodology and power calculations in greater detail; (2) added a supplemental figure demonstrating the relationship with the MemAID clinical trial; (3) added a supplemental figure showing the trajectory of depressive symptoms longitudinally; and (4) addressed formatting concerns. The following paragraphs detail the specific changes in response to each editor and reviewer comment.

Editor’s Comments:

Thank you for submitting your manuscript to PLOS ONE. After careful consideration, we feel that it has merit but does not fully meet PLOS ONE’s publication criteria as it currently stands. Therefore, we invite you to submit a revised version of the manuscript that addresses the points raised during the review process.

Remark: Some formatting amendments are mandatory.

 Please detail the methodology as appropriate.

Thank you for the opportunity to revise and resubmit the manuscript! We have incorporated the formatting amendments and clarified our research methods in our revised manuscript. 

We have edited out manuscript to ensure that it meets PLOS ONE’s style requirements, including those for file naming.

"S.S.B. served as a consultant for Kinto Care from 2019-2020. L.N. provided consultation to the Radiological Society; to the Journal of Cardiovascular Magnetic Resonance; to Five Island Consulting LLC, Georgetown ME; and to Vinmec Inc. Hanoi, Vietnam between 2015 and 2020. The authors report no conflicts with any product mentioned or concept discussed in this article."

As requested, we would like to modify our Competing Interests section as follows:

S.S.B. served as a consultant for Kinto Care from 2019-2020. L.N. provided consultation to the Radiological Society; to the Journal of Cardiovascular Magnetic Resonance; to Five Island Consulting LLC, Georgetown ME; and to Vinmec Inc. Hanoi, Vietnam between 2015 and 2020. This does not alter our adherence to PLOS ONE policies on sharing data and materials.

The dataset from this study is property of Beth Israel Deaconess Medical Center (BIDMC) and contains sensitive medical information including information about past and current treatment of psychiatric disorders. In order to provide access to deidentified data from this dataset, BIDMC will require a data sharing agreement with a requesting investigator’s institution. Additionally, the BIDMC IRB would require acknowledgment that the receiver has obtained an exemption from their local ethics/IRB committee that any shared data is exempt from Human Subject Research. Therefore, the data can be accessed by reaching out to the Research Administrator of the BIDMC Neurology Department, who would then put the necessary agreements in place to facilitate data sharing. 

Contact Name: Stacy Mueller

Title: Research Administrator, Neurology Department, BIDMC

Email: slmuelle@bidmc.harvard.edu

Phone: 617-667-1984

Address: Beth Israel Deaconess Medical Center, 330 Brookline Ave, Boston MA

As requested, we have included captions for our supporting information files at the end of the manuscript and edited the manuscript to adhere to PLOS ONE’s Supporting Information guidelines.

We have reviewed our reference list and it is complete. Based on reviewer comments, we have edited the Introduction and Methods clarify the study rationale and improve reproducibility. As such, some of the references have changed from the initial submission.

We have removed references from:

-Lutz, et al (old 1)

-Rowe, et al (old 2)

We have added references from:

-United States Census Bureau (new 1)

-Bauer, et al (new 3)

-Vaughan, et al (new 10)

-Morin, et al (new 12)

-NIH (new 24)

Reviewer’s Comments:

Dear authors! Thank you for conducting this study, which might also assist mental health professionals in recognizing and treating depression symptoms in the elderly population to reduce disabilities related to depression symptoms. However, your methods and materials weren't quite clear.

Thank you for your overall enthusiasm as well as your time and effort in reviewing this manuscript! We appreciate your help in identifying areas of our materials and methods sections which were not clearly described and have revised the manuscript accordingly. After incorporating your comments we feel that our revised manuscript is stronger, more concise, and has improved reproducibility.

Abstract: Has all scientific information and it is clearly written, but:

Line 29-31 –You have been prospectively analyzed data from 223 adults (age 50-85) over 48 weeks who were participating in a clinical trial “Memory Advancement by Intranasal Insulin in Type 2 Diabetes. But in this manuscript, line 89 - only 106 study participants were type2 Diabetes mellitus patients (T2DM). these two phrase contradict to each other , it needs correction 

The MemAID trial, from which this data is drawn, included both participants with type-2 diabetes and controls without type-2 diabetes. We have added clarifying information to the abstract:

- Page 2; Line 50-51: We prospectively analyzed data from 223 adults (age 50-85; 117 controls and 106 with type-2 diabetes) over 48 weeks who were participating in a clinical trial “Memory Advancement by Intranasal Insulin in Type 2 Diabetes.”

In the abstract part, you stated that study participants' cognition, walking speed, and medical disabilities were assessed at baseline, but it is not clear how and by what medical diagnostic workup, cognition, walking, and medical comorbidities were assessed. Furthermore, what about study participants' status regarding their baseline cognition, walking speed, and medical comorbidities? I think all these need a brief and clear description.

We have added a brief and clear description of the assessments for cognition, walking speed, and medical comorbidities to the Abstract:

- Page 2; Line 52-56: Data from self-reported disability (World Health Organization Disability Assessment Schedule) and depressive symptoms (Geriatric Depression Scale) were obtained from baseline, week 25, and week 48 visits. Cognition (Mini-mental status examination) and medical comorbidities (Charlson Comorbidity Index) were assessed at baseline.

- Page 2; Line 60-61: At baseline, depressive symptoms, cognition, and walking speed were within normal limits, but participants had a high 10-year risk of cardiovascular mortality.

Line 31& 32 - The World Health Organization Disability Assessment Schedule 2.0 (WHODAS) measured disability. And line 33 & 34 -WHODAS were assessed at Baseline and at 8-week intervals - it seems like redundancy and need revision in one line 

We have edited the abstract to avoid redundancy:

- Page 2; Line 52-54: Data from self-reported disability (World Health Organization Disability Assessment Schedule) and depressive symptoms (Geriatric Depression Scale) were obtained from baseline, week 25, and week 48 visits.

Better to revise and rewrite Key words, considering MeSH (Medical subject headings) terms rather than repeating words in introduction.

Key words have been revised using MeSH terms:

- Depression; International Classification of Functioning, Disability and Health; Aging; Mediation Analysis; Cognition; Gait Analysis

You have been investigated mood as an important modifiable risk factors that my mediate the effect of physical and cognitive decline on disability. But, it is obvious that mood is a broad term which can be characterized by individual’s subjective feelings that might be normal, depressed, irritable, and expansive and etc. And sometimes it can be a gate symptom for manic episode. So it is beyond your study title. The researchers should justify this fact, why they were used broad term “mood” as important modifiable risk factors. 

We agree that the term “mood” is too broad and does not accurately describe our study’s scope. Therefore, we have changed the term “mood” to “depressive symptoms” throughout the manuscript where applicable.

Introduction 

Line 54 -what is the recent global total estimated number of your study population (50–85-years-old)? Furthermore, it is preferable to describe the total number of elderly people in your study area/setting.

We agree that it is best to focus on demographics related to the study population. Therefore, we have edited the Introduction to include the estimated population of adults over the age of 50 in the United States:

- Page 3, Line 128-129: The population of adults over the age of 50 currently makes up 35 percent of the US population…

It would be better if the introduction part of this study had more focus and described the magnitude of depression symptoms' impact on disabilities among the elderly population.

We thank the reviewer for this feedback and we have edited the Introduction to include a greater focus on depressive symptoms and the impact on disability on older adults:

Page 3, Line 135-142: Depressive symptoms are found in 11-25.3 percent of older adults and are independently associated with disability. In particular, anxiety and somatic symptoms related to depression are linked with a greater risk of disability in older adults. Furthermore, depression may exacerbate disability when comorbid with other conditions such as chronic pain. However, it remains unknown whether depressive symptoms prospectively mediate the relationship between cognition, mobility, medical risk factors and disability. 

Line -55 -is not complete sentences and does not give sense

This sentence has been edited for clarity:

- Page 3, Line 130-131: The prevention of cognitive, physical, and medical disability in older adults is one important component of promoting active and independent living in older adults.

Methods and materials 

The section on ‘methods and materials’ in this manuscript weren’t written clearly or appropriately. As a result, it generally needs to be revised and rewritten in order to be understandable and scientifically sound. For instance, in “Study setting and design” part, line 79- Data for the present analysis were collected before and after the treatment period of intranasal insulin. Which is not appropriate and right way to write about data collection procedure in “ study setting and design” section 

We thank the reviewer for this feedback. We have revised and re-written the Methods and Materials Section and feel this has greatly improved the description of study procedures. We now have a separate section for “Study Design:” 

- Page 4-5, Line 179-210: All participants in this prospective cohort were concurrently enrolled in the MemAID study, a randomized, double-blinded, placebo-controlled clinical trial (ClinicalTrials.gov NCT02415556, FDA IND 107690). All study procedures occurred between October 6, 2015 and May 31, 2020. MemAID methodology has been previously published; S1 Fig shows the relationship of the present dataset to the overall MemAID trial. Briefly, study participants completed screening and baseline assessments prior to initiation of the study drug or placebo. These included a comprehensive medical history, physical examination, blood draw, MMSE, Geriatric Depression Scale (GDS), 36-item World Health Organization Disability Assessment Schedule 2.0 (WHODAS) questionnaire administration, and gait assessment. Participants were asked to self-report demographic information including sex (Male or Female), race (American Indian or Alaska Native, Asian, Black or African American, Native Hawaiian or Other Pacific Islander, White, or more than one race), ethnicity (Hispanic or Latino or Not Hispanic or Latino), and years of education (total years including graduate education if applicable). Participants then attended nine study visits over 48 weeks, which included assessment of GDS, WHODAS, and gait. In order to minimize confounding from any treatment effect, data for the present analysis was used only from time-points before or after study drug administration.

I note that the Ethics approval information you provided in the ‘Participants section’, line 83-84 is not appropriate. Better if the researchers provide separate headings for ‘Ethics approval’ 

As requested, we have added a separate heading for “Ethics Approval” (Page 4, Line 173).

Line 141-142, the primary outcome was WHODAS and exposures of interest were GDS, NW, MMSE, and CCI. WHODAS, GDS, MMSE, and CCI have been used to collect data related to dependent and independent variables/outcomes. So, WHODAS, GDS, MMSE, and CCI cannot be considered as primary out-comes. The author of this is study expected to justify this issue.

We thank the reviewer for pointing out that defining “outcome” variables may not be relevant for this prospective cohort study. We have clarified the description of our statistical analyses: Our dependent variable of interest is WHODAS, which assays self-reported clinical disability. Our independent variables of interest are MMSE, NW, CCI. GDS is an additional independent variable of interest which is investigated as a mediator in our analyses. We have edited the manuscript to be consistent in statistical terminology:

Page 8, Line 300-302: The clinical dependent variable of interest was WHODAS. The independent variables of interest were NW, MMSE, and CCI, and the independent variable investigated as a mediator was GDS.

Line 102-103 -The primary outcomes were measures of disability. What kind of disability? Is it physical, cognitive or other types of disability? it is not clear 

The clinical dependent variable of interest is WHODAS, a measure of disability which includes physical, cognitive, and social components. A description of WHODAS can be found:

- Page 6, Line 238-240: The WHODAS 2.0 is a self-reported measure of disability. The WHODAS quantifies difficulty preforming activities of daily living in six functional domains: cognition, mobility, self-care, getting along with people, life activities, and participation in society.

“Sample size” written next to “Post-hoc T2DM Subgroup Analysis”. This also not appropriate. Sample Size Determination and Sampling Technique followed are not clear, better if clearly described.

At the reviewer’s suggestion we have moved our sample size calculation to the “Statistical analysis” section, since the calculation was done for the primary analysis. Our primary power analysis tested our ability to have sufficient statistical power to detect an association between the independent variables of interest (MMSE, NW, CCI, and GDS) and the dependent variable (WHODAS). We have re-drafted the description of the power analysis to make add greater detail and clarity, and have also added a power analysis of the longitudinal models:

- Page 8, Line 307-314: With a power at 0.8 or higher and a type 1 error set at 0.05, our sample of 223 participants in the cross-sectional analysis allows us to detect an effect size of r=0.19 (correlation coefficient; a moderate association) for the association between each of the three independent variables of interest (MMSE, NW, CCI) and the dependent variable of interest (WHODAS).

In the longitudinal analysis, our sample of 155 participants allows us to detect an effect size of r=0.22 (a moderate association) for the association between each of the three independent variables of interest (MMSE, NW, CCI) and the dependent variable of interest (WHODAS) with a power at 0.8 or higher and a type 1 error set at 0.05.

When this study was employed? Better if study period will be incorporated to the methodology section 

Study dates were added to the Study design section of the Methods:

- Page 4, Line 181-182: All study procedures occurred between October 6, 2015 and May 31, 2020.

All the following subheadings, better to rewrite in sentence case:

Line 73 Materials and Methods can be correct as Methods and materials 

Line 74 Design and Setting can be correct as Study setting and design 

Line 106 Assessment of Disability and Functionality can be corrected as Assessment of disability and functionality

Line 116 Assessments of Mood and Cognitive Function can be corrected as Assessments of mood and cognitive function

Line 124 Assessment of Mobility can be corrected as Assessment of mobility

Line130 Assessment of Medical Comorbidities and 10-year Mortality Risk can be corrected as Assessment of medical comorbidities and 10-year mortality risk 

 Line 140 Statistical Analyses can be corrected as Statistical analyses

Line 146 Cross-sectional Analyses can be corrected as Cross-sectional analyses 

Line 157 Longitudinal Analysis 157 of Change in Disability Can be corrected as Longitudinal analysis 157 of change in disability

Line 173 Longitudinal Mediation Analyses 

Post-hoc T2DM Subgroup Analysis 

Line 203 Sample Size 

Line 209 Demographic and Clinical Variables

Per reviewer feedback, all titles have been changed to sentence case.

Your study design is not clear and not in line with your study objective / title –it is stated as Participants were enrolled in the Memory Advancement with Intranasal Insulin (MemAID) study, a randomized, double-blinded, placebo-controlled clinical trial (ClinicalTrials.gov NCT02415556, FDA IND 107690). 

We appreciate the reviewer pointing out the need for greater clarity in the study design. Data for this prospective cohort was collected from participants concurrently enrolled in the MemAID trial of intranasal insulin. For cross-sectional analyses we only used baseline data (before treatment), and for longitudinal analyses we used data from week 25 and week 48 (after the treatment period), thus reducing potential confounding effects from the trial. To clarify we have added additional details to the Study design section in Methods (Page 4-5, Line 179-210). We have also added S1 Fig, which demonstrates the relationship between the prospective cohort and the MemAID trial:

- S1 Fig and Page 28, Line 880-886: S1 Figure. Study Flow Diagram. All participants included in the present study were concurrently enrolled in the MemAID clinical trial of intranasal insulin. Only data included in the longitudinal analysis are shown and were drawn from visits at baseline, 25 weeks, and 48 weeks. Additional assessments which were performed as part of the MemAID trial have been previously published (Novak, et. Al., Journal of Neurology 2022) and are not shown in the figure. INI: intranasal insulin; MMSE: Mini mental state examination; WHODAS: World Health Organization Disability Assessment Schedule 2.0; GDS: Geriatric Depression Scale.

At the beginning why study participants were enrolled in the Memory Advancement with Intranasal Insulin problems, did they have known diagnosed Memory problems? If so, better to describe briefly. 

Participants did not have any diagnosed memory problems, and were excluded if they had a diagnosis of dementia or an MMSE score <20. This is described in the Participants section of Methods and Materials:

- Page 5, Line 214-215: Eligible participants were 50 to 85 years old, with or without type-2 diabetes (T2DM), able to walk for six minutes, had a Mini-Mental State Examination (MMSE) >20 with no diagnosed dementia…

The study design for your study title is not clear enough. Is it Crossectional, case-control RTC or prospective cohort study? It is not clear, because the following statements are stated as in - Line 79-80 –‘Data for the present analysis were collected before and after the treatment period of intranasal insulin, Line 88-89 -For the cross-sectional analysis, we used data from 223 participants enrolled in the MemAID trial (117 controls and 106 with T2DM) and Line-96-97-One participant who completed the MemAID trial did not 97 complete weeks 48 WHODAS so was excluded from the present analysis. Why one participant only excluded from your study and what about those 67 participants who have not been included in longitudinal analysis in you study? Better to justify 

We appreciate the reviewer’s comments that the study design was not clear and we have made changes throughout the manuscript to clarify the study design. This is a prospective cohort study which was embedded within the MemAID trial of intranasal insulin. To clarify the study design we edited the title to “Depressive symptoms exacerbate disability in older adults: A prospective cohort analysis of participants in the MemAID trial (Page 1, Line 1-2). We added additional details added Study design section in Methods, which goes into greater detail about the relationship with the MemAID trial (Page 4-5, Line 179-210). We added a S1 Fig to further describe the relationship between this prospective cohort study and the MemAID trial (S1 Fig and Page 28, Line 880-88). Finally, we edited the Fig 1 Legend (CONSORT diagram) to explain the reasons that data from 67 participants were not available at follow up:

- Page 6, Line 230-235: All 223 participants who completed screening and baseline assessments were included in the cross-sectional analyses. Of these, 66 did not complete the MemAID study (either withdrew from the study, were terminated by the investigator, or were lost to follow-up), and one completed the MemAID study but was missing data for the week 48 WHODAS. Therefore, the remaining 155 participants were included in the longitudinal analysis of this prospective cohort.

At the beginning you stated that study participants were evaluated at base line, then at the intervals of every 8-weeks (i.e., 8, 16, 24, 32, 40 and 48 weeks of their participation in MemAID ) , but you did not stated/showed their evaluation of depression symptoms , cognitive , walking status and their disability status at each of these evaluation periods . It needs clear and brief description of participants evaluation results at 8, 16, 24, 32 , 40 and 48 weeks 

We thank the reviewer for pointing out a need for greater clarity. This study used only data points from baseline study visits (before treatment) and from week 25 and week 48 (after the treatment period) to minimize confounding effects. We have added S1 Fig to show the data collection timeline for the present study, and relationship to the MemAID study procedures, which have been previously published (Novak, et. Al., Journal of Neurology 2022).

Some of your study results stated within Methods and materials section. For instance, line 153 to 154, since 78% of participants self-reported race as “White,” race was encoded as a dichotomous variable (White vs. Non-White).

These results have been moved to the Results section (Page 11, line 409-410).

Result 

The first column of Table 1, line 225, is unclear; are they variables? If so, categorization is recommended.

Table 1 shows demographic and clinical variables of all participants at Baseline. We have added a column heading to Table 1 (Page 11) “Demographics and clinical variables at baseline” to improve clarity.

What are the differences between the study participants' race and ethnicity classification as shown in Table 1?

For this study we asked participants for self-reported race and ethnicity, following the NIH’s recommended Race and Ethnicity categories NOT-OD-15-089: American Indian or Alaska Native, Asian, Black or African American, Native Hawaiian or Other Pacific Islander, and White. Participants could also self-identify as More than one race. There are two categories for ethnicity: Hispanic/Latino and Not Hispanic/Latino. A reference to the NIH guidelines has been added to the manuscript and a description was added to the “Study design” section:

- Page 5, Line 202-207: Participants were asked to self-report demographic information including sex (Male or Female), race (American Indian or Alaska Native, Asian, Black or African American, Native Hawaiian or Other Pacific Islander, White, or more than one race), ethnicity (Hispanic or Latino or Not Hispanic or Latino), and years of education (total years including graduate education if applicable).

For clarification, some of the variables in column one of Table 1—including sex, ethnicity, years of education, and others—need further categorization.

Self-reported sex was recorded as ether Male or Female. Years of education were recorded as number of years attending school. This information has been added to “Study design” (Page 5, Line 202-207) and Table 1 Legend:

- Page 12, Line 435-436: Demographics shown include self-reported variables of sex (male/female), race and ethnicity, and years of education.

Table 1 shows the agea, (%) female) b, (% Latinx) b, and other variables. What are a and b represented in this table's column 1? Better to write Key notes under the table 

These superscripts denote the statistical test to use to test for between-group differences of each variable. Per the reviewers note we have added moved the description of the statistical tests from the Table 1 Legend to a new Table 1 Key for greater readability:

- Page 12, Line 429-430: aPooled 2-tailed t-test assuming equal variance b2-Tail Fisher’s Exact test cPearson’s Chi-square test

According to your study 175 participants did not show depressive symptoms at baseline. During your longitudinal depression evaluation by using GDS, how many of them developed depressive symptoms over 48 week’s period? 

We appreciate the reviewer’s question about how GDS changes over the course of the study. We have added an analysis of longitudinal GDS, which showed relative stability of group level means, but significant individual variability in trajectory of mood over time:

- Page 14, Line 491-492: At the group level, there was no significant difference in GDS at baseline, Mid-Study, or 48 weeks (S2 Fig). However, while mean WHODAS and GDS were relatively stable, there was clinically meaningful variability within individual participants during the course of the study in WHODAS Change (mean=-1.3, S.D.=8.5, range -42.3 to 31.5) and GDS Change (mean=-0.45, S.D.=4.0, range -12 to 21). Of the participants without depressive symptoms at baseline, 12 developed depressive symptoms by Mid-Study (GDS>10). Overall during the course of the trial, 46 participants experienced both worsening GDS and worsening WHODAS, while 24 participants experienced both improving GDS and improving WHODAS.

- S2 Fig and Page 28, line 888-890: S2 Figure. Longitudinal depressive symptoms. Mean depressive symptoms over the duration of the study are shown. At the group level, depressive symptoms were stable between baseline, Mid-study (week 25), and end of the study (week 48). GDS: Geriatric Depression Scale.

Discussion 

The authors discussed their study findings in depth which is scientifically appropriate and smart. It is better if the researchers provide separate headings and rewrite for “abbreviations and availability of data and materials “next to conclusions section 

We appreciate the reviewer’s positive comments about our discussion section! We have added Abbreviations and Availability of Data and Materials sections (Page 23-24, Line 717-738).

---

## [Editor Report · Decision Letter 1]

15 Nov 2022

Depressive symptoms exacerbate disability in older adults: A prospective cohort analysis of participants in the MemAID trial

PONE-D-22-17003R1

Dear Dr. Buss,

We’re pleased to inform you that your manuscript has been judged scientifically suitable for publication and will be formally accepted for publication once it meets all outstanding technical requirements.

Kind regards,

Lai Kuan Lee

Academic Editor

PLOS ONE
---

## [Editor Report · Acceptance letter]

17 Nov 2022

PONE-D-22-17003R1 

Depressive symptoms exacerbate disability in older adults: A prospective cohort analysis of participants in the MemAID trial 

Dear Dr. Buss:

I'm pleased to inform you that your manuscript has been deemed suitable for publication in PLOS ONE. Congratulations! Your manuscript is now with our production department. 

Kind regards, 

on behalf of

Dr. Lai Kuan Lee 

Academic Editor

PLOS ONE